# Management of Low Back Pain: Do Physiotherapists Know the Evidence-Based Guidelines?

**DOI:** 10.3390/ijerph20095611

**Published:** 2023-04-23

**Authors:** Antoine Fourré, Rob Vanderstraeten, Laurence Ris, Hilde Bastiaens, Jozef Michielsen, Christophe Demoulin, Ben Darlow, Nathalie Roussel

**Affiliations:** 1Department of Neurosciences, Research Institute for Health Sciences and Technology, University of Mons, 7000 Mons, Belgium; 2Faculty of Medicine and Health Sciences, Rehabilitation Sciences and Physiotherapy (MOVANT), University of Antwerp, 2610 Antwerpen, Belgium; 3Orthopedic Department, University Hospital, 2650 Antwerp, Belgium; 4Department of Sport and Rehabilitation Sciences, University of Liège, 4000 Liège, Belgium; 5Department of Primary Health Care and General Practice, University of Otago, Wellington 6021, New Zealand

**Keywords:** red flags, adherence, beliefs, knowledge, attitudes

## Abstract

Background: Clinical practice guidelines promote bio-psychosocial management of patients suffering from low back pain (LBP). The objective of this study was to examine the current knowledge, attitudes and beliefs of physiotherapists about a guideline-adherent approach to LBP and to assess the ability of physiotherapists to recognise signs of a specific LBP in a clinical vignette. Methods: Physiotherapists were recruited to participate in an online study. They were asked to indicate whether they were familiar with evidence-based guidelines and then to fill in the Health Care Providers’ Pain and Impairment Relationship Scale (HC-PAIRS), Back Pain Attitudes Questionnaire (Back-PAQ), Neurophysiology of Pain Questionnaire (NPQ), as well as questions related to two clinical vignettes. Results: In total, 527 physiotherapists participated in this study. Only 38% reported being familiar with guidelines for the management of LBP. Sixty-three percent of the physiotherapists gave guideline-inconsistent recommendations regarding work. Only half of the physiotherapists recognised the signs of a specific LBP. Conclusions: The high proportion of physiotherapists unfamiliar with guidelines and demonstrating attitudes and beliefs not in line with evidence-based management of LBP is concerning. It is crucial to develop efficient strategies to enhance knowledge of guidelines among physiotherapists and increase their implementation in clinical practice.

## 1. Introduction

The leading cause of disability worldwide is low back pain (LBP) [1,2]. All clinical guidelines for the management of LBP recommend diagnostic triage to differentiate LBP presentations into those with features of underlying serious pathology (such as infection or cancer), those with features of specific LBP (such as radiculopathy or spinal stenosis) and those with non-specific LBP [3,4,5,6]. However, there is a lack of studies exploring the ability of first-line healthcare practitioners (HCPs) to suspect the presence of an underlying pathology. However, although most patients suffer from non-specific LBP, the ability to recognize the possibility of serious spinal pathologies is crucial as the management of patients with specific LBP will be completely different [6].

Although non-specific LBP is explained by a combination of biological, psychological and social factors, many HCPs still consider LBP to be the result of one single (biomedical) factor [2] and focus care on this biomedical factor. However, clinical guidelines underline the importance of evaluating psychosocial factors as these could lead to an increased risk of chronicity [7,8,9]. 

Optimal management of patients with non-specific LBP includes explanation, reassurance, promotion of movement, return to work and self-management. However, many HCPs, especially those with a biomedical orientation [10], do not follow these recommendations [10,11,12] and manage patients with LBP in a guideline-inconsistent way. This approach is associated with increased use of diagnostic imaging, opioids, spinal injections and surgery, contributing to persistent disability and enormous costs for society [1,13,14].

Therefore, the objectives of this study were (1) to question physiotherapists about their knowledge of evidence-based guidelines for the management of LBP and their application in clinical practice; (2) to examine their knowledge, attitudes and beliefs concerning LBP and the association with their self-reported knowledge of the guidelines; and (3) to assess their recommendations about activity and work and their ability to suspect or detect a specific cause of LBP in a clinical vignette.

## 2. Materials and Methods

### 2.1. Design

This cross-sectional study reports baseline assessment from a randomised controlled study registered on clinicaltrials.gov (NCT05284669). The study was approved by the local ethical committee. The results of this study are reported using the STROBE guidelines for observational studies [15].

### 2.2. Setting

This study was carried out using an online setting. Participants accessed an internet platform (https://qualtrics.com) (accessed on 10 September 2022) detailing study information using their own internet device (e.g., computer, tablet or smartphone). After providing informed consent, participants were invited to complete the online survey.

### 2.3. Participants

Licensed Dutch- and French-speaking physiotherapists in Belgium and France were informed about the possibility to participate in an online study. Various strategies were used [16] to contact clinically active physiotherapists in Belgium and France. Invitations were shared in two languages (Dutch and French) in broad networks such as national associations (e.g., Axxon, Domus Medica, etc.), local networks of university departments and hospitals, registered physiotherapy associations, etc. Eligibility criteria were French-speaking or Dutch-speaking graduated physiotherapists working in Belgium or France. Exclusion criteria were no management of patients with low back pain and not being in possession of an internet-connected device. Recruitment took place between August 2021 and December 2021.

### 2.4. Outcomes

This study included five questionnaires: a self-developed socio-demographic questionnaire, the Health Care Providers’ Pain and Impairment Relationship Scale (HC-PAIRS) [17,18], the 10-item version of the Back Pain Attitudes Questionnaire (Back-PAQ-10) [19,20], the revised Neurophysiology of Pain Questionnaire (NPQ) [21,22] and questions relating to two clinical vignettes (one about a patient with non-specific LBP [23] and one about a patient with a specific LBP). All questionnaires were available in the language of the participant (either French or Dutch). The Back-PAQ and the NPQ were translated into Dutch using a back-and-forth translation process using Beaton’s guideline with four translators (two French-speaking and two Dutch-speaking) [24]. The HC-PAIRS and the vignette (non-specific LBP), translated in a previous study with the same process, were used for the French-speaking participants [25].

#### 2.4.1. Sociodemographic

This questionnaire was developed for this study. It included several items related to personal factors (age, gender, region, clinical occupation and settings) of participants. Two questions (Yes or No answer) were asked; the first was about the confidence in their own knowledge of guidelines for the management of LBP, and the second was about their application of guidelines in clinical practice.

#### 2.4.2. Health Care Providers’ Pain and Impairment Relationship Scale (HC-PAIRS)

The HC-PAIRS assesses attitudes and beliefs concerning physical impairments for patients with chronic LBP [26]. It consists of 13 statements that are rated on a seven-point Likert scale ranging from ‘totally disagree’ to ‘totally agree’. The total score ranges from 13 to 91. A high score on the HC-PAIRS reflects a belief in a strong relationship between pain and impairment [17]. The good psychometric properties of this questionnaire have been established in graduated HCPs, including physiotherapists [17,26,27].

#### 2.4.3. Back Pain and Attitudes Questionnaire (Back-PAQ)

The Back-PAQ questionnaire (10-items version) [19] assesses attitudes and underlying beliefs about back pain on a five-point Likert scale. The scoring of the answers ranges from +2 to −2. Items 6-7-8 have a reversed score. The total score ranges from −20 to 20. A negative score reflects beliefs that are unhelpful and vice-versa. To interpret the Back-PAQ, five themes are related to the items: “the vulnerability of the back”, “the relationship between back pain and injury”, activity participation while experiencing back pain”, “psychological influences on recovery” and “the prognosis of back pain”. All items were written in the second person to personalise the questionnaire. The purpose of this personalisation is that responders present their own beliefs rather than projecting their beliefs onto people with LBP or presenting their beliefs about people with LBP [19]. 

#### 2.4.4. Neurophysiology of Pain Questionnaire (NPQ)

The Neurophysiology of Pain Questionnaire (NPQ) assesses how an individual conceptualises biological mechanisms underpinning pain [21]. The NPQ includes 19 questions with three response options (true; undecided; false). The scoring is 1 for a correct answer and 0 for a wrong or undecided answer. Higher scores reflect better knowledge of the pain neurophysiology. This questionnaire was included to evaluate if physiotherapists accurately understand the neurophysiology of pain [22,28] as pain education could improve kinesiophobia and pain catastrophising in patients with chronic LBP [29].

#### 2.4.5. Clinical Vignettes

Two clinical vignettes were used in this study. The first vignette was one of the three vignettes developed by Rainville et al. [23]. It describes a patient with non-specific LBP. The participant was asked to give his/her opinion on the appropriate level of activity for the patient, with choices ranging from 1 (no limitations on activity) to 5 (limit all physical activity) and assess the patient’s ability to work from 1 (full-time) to 5 (remain out of work). If the score of the participant was between 1 and 2, it was considered guideline-consistent [25]. If the score was between 3 and 5, it was considered guideline-inconsistent [25]. The total score ranging from 2 to 10 was calculated using the sum of the two items.

A second vignette was developed to analyse the capacity of physiotherapists to suspect a specific underlying spinal pathology (i.e., to evaluate the skills of the diagnostic triage) and describe the symptoms of a patient with a specific cause of LBP (lumbar spinal stenosis). The methodology of Jette et al. was used to develop this vignette [30]. Participants answered an open question: “In your opinion, what are the causes/contributing factors to the pain of this patient?” Answers of the participants were scored on two criteria: “ability to suspect a specific LBP” and “ability to detect the correct specific LBP”. Participants were scored 1 (“yes”) if they suspected or detected the specific LBP in the vignette and 0 (“no”) if they did not.

### 2.5. Statistical Methods

Data were downloaded from Qualtrics and sorted using Microsoft Excel (16.57) (Microsoft Corporation, Redmond, WA, USA). IBM Statistics 28 (IBM SPSS, Armonk, NY, USA) was used to perform statistical analyses. Only participants with complete data (i.e., all questionnaires completed) were included in the statistical analyses. 

Descriptive statistics were used for all the questionnaires and vignettes. Normality tests of outcomes results were performed (Kolmogorov–Smirnov Test). Kruskall–Wallis and Mann–Whitney tests with a significance of 0.05 were used to compare the total score of the questionnaires with the knowledge of the guidelines, groups of physiotherapists seeing less (<15) or more (15–20) patients with LBP per month and the ability to suspect or detect the specific diagnosis of LBP. Both vignettes were analysed using descriptive statistics to determine the number of physiotherapists giving guideline-inconsistent recommendations and being able to suspect or detect a specific cause of LBP.

## 3. Results

In total, 2447 HCPs opened the questionnaire online. After exclusion of participants (see Figure 1), 527 physiotherapists from two countries (59% females and 41% males; see Table 1) were included in the data analysis. 

Their clinical occupation was mainly full-time (81%). Two-thirds of the physiotherapists (63%) reported seeing at least 10 new patients with LBP per month. The majority (63%) of the physiotherapists reported they were uncertain or did not know the content of guidelines on the management of LBP and only 31% reported applying them in clinical practice.

### 3.1. Knowledge, Attitudes and Beliefs of Physiotherapists

Descriptive statistics are detailed in Table 2. No significant differences were found in the scores of the questionnaires between physiotherapists seeing less (<5) or more (15–20) patients with LBP per month except for the Back-PAQ (*p* = 0.02). No significant differences were found between Belgium and France for these questionnaires (data not shown).

The results of the Back-PAQ were analysed by theme and are detailed in Table 3. The worst scores were related to the theme “vulnerability of the back”, with 43% of physiotherapists having neutral or negative beliefs.

Physiotherapists were sub-grouped based on the self-reported knowledge of guidelines for the management of LBP. The scores of participants reporting that they know the guidelines were significantly better (i.e., more guideline-consistent) for the HC-PAIRS, Back-PAQ and NPQ (*p* < 0.001) compared to those who reported to be unfamiliar with them (see Figure 2).

### 3.2. Vignettes

The descriptive results of the vignette describing a patient with non-specific LBP are presented in Table 4. Most of the physiotherapists (63%) gave guideline-inconsistent recommendations for work. Concerning activity, 24% of the physiotherapists gave guideline-inconsistent recommendations.

A significant difference between the self-reported knowledge of the guidelines and the vignette’s total score was found (*p* = 0.009). No significant difference was found between the self-reported application of the guidelines and the vignette’s total score (*p* = 0.079).

The descriptive results of the vignette describing a case with specific LBP are presented in Table 5. Fifty-four percent of the physiotherapists suspected the presence of a specific underlying cause of LBP in this vignette and only 30% of them mentioned the correct spinal pathology. Participants who suspected the presence of a specific cause of LBP had significantly better scores in the NPQ (*p* = 0.037). Participants who detected the specific cause of LBP (spinal stenosis) had significantly better scores in the Back-PAQ and NPQ (*p* = 0.004).

## 4. Discussion

The results of this study revealed that a low proportion of physiotherapists in Belgium and France report knowing or using LBP guidelines. Physiotherapists not familiar with the guidelines were more likely to have attitudes indicating a strong relationship between pain and impairment, beliefs about LBP that are unhelpful, inadequate knowledge on the neurophysiology of pain and guideline-inconsistent recommendations regarding work. Half of the physiotherapists in this study did not suspect a specific cause of LBP in a clinical vignette.

### 4.1. Physiotherapy in Belgium and France

In both countries, patients need a referral prescription from a physician to have access to physiotherapy and to be reimbursed by the health social security system [31]. In Belgium, the number of sessions is limited to 18 sessions. Direct access to physiotherapy is not yet implemented in Belgium but an experimental study is currently performed to evaluate the (cost-)effectiveness of direct access [32]. In France, direct access is allowed for specific cases (acute LBP and ankle sprain in multidisciplinary centers) but it is not widely implemented. The results of this study found that knowledge, attitudes and beliefs of physiotherapists are equivalent in Belgium and France.

### 4.2. Knowledge of the Guidelines and Questionnaire Scores (HC-PAIRS, Back-PAQ and NPQ)

The low proportion of physiotherapists reporting to know guidelines for the management of LBP is striking as significantly more guideline-inconsistent attitudes and behaviours (i.e., reflected by significantly worse scores on HC-PAIRS, Back-PAQ, NPQ, recommendations based on clinical vignette) were observed in physiotherapists uncertain of or not knowing clinical guidelines. This proportion is significantly higher compared to a study in Australia where only 19% of physiotherapists were uncertain of clinical guideline recommendations [33]. These differences might be explained by a combination of reasons: undergraduate education [34], promotion of guidelines (media campaigns, professional bodies, insurance/funding) [35], health system design [36,37] and a cultural shift toward evidence-based care.

However, our results are in line with previous studies; one reported that only 12% of physiotherapists were aware of clinical guideline recommendations [38], and another one reported that only 52% of physiotherapists used guidelines in clinical practice [39]. A systematic review found that physiotherapists questioned the relevance of guideline recommendations (such as assessing cognitive, psychological and social factors of patients) or felt they had inadequate clinical skills [40]. A recent study found that the proportion of physiotherapists providing guideline-recommended treatment is still low and has not increased since 1990 [41]. It is relevant to report that a high proportion of physiotherapists (30%) in this study were working as solo practitioners. Working in isolation could have an impact on the development of clinical expertise and implementation of evidence-based care. These results are highly concerning and reveal the urgent need to develop better strategies to implement evidence-based guidelines.

Concerning the HC-PAIRS, recent studies using the 13-item version in physiotherapists in the USA [42] and New Zealand [27] found lower scores (i.e., median of 31 compared to 42 in our study), suggesting a more bio-psychosocial orientation of participants in these countries. Higher scores on the HC-PAIRS are not only associated with a more biomedical treatment orientation, but this can also negatively influence health attitudes and behaviour of the patients [43]. It is known that HCPs’ beliefs about LBP might be associated with the beliefs of their patients [44]. While this study did not investigate the effective management of physiotherapists during actual consultations, the high scores of the HCPs who were clinically active physiotherapists are nevertheless concerning as it might suggest that these physiotherapists provide predominantly biomedical management to their patients. Self-reflection strategies should be implemented in the education of physiotherapists to understand how their beliefs about pain align with evidence and the negative effects that biomedically focused care can have on patient outcomes [45].

The short version of the Back-PAQ (10-item) with −2 to +2 scoring was chosen in this study to facilitate the interpretation of results. Negative scores represent beliefs that are not helpful concerning LBP. Physiotherapists in this study seeing more patients (15–20) per month had significantly better beliefs concerning LBP compared to those seeing less (< 5). The clinical expertise of physiotherapists working with more patients with LBP could have influenced this result, but it is important to note that no difference was observed for the other questionnaires (HC-PAIRS and NPQ). On average, physiotherapists presented positive scores, meaning they have beliefs more aligned with helping recovery. Similar results were found in recent studies [46,47]. Nevertheless, there was a lot of room for improvement. The analysis of the Back-PAQ themes showed that 43% of physiotherapists had negative or neutral answers concerning the items related to the vulnerability of the back. This means that many physiotherapists believed that is easy to injure the back and that caution is needed. These beliefs related to the need of protection reflect guideline-inconsistent beliefs related to the biomedical model. In other countries, some studies presented lower (worse) scores for the Back-PAQ in physiotherapists [27,48]. These results highlight the urgent need to develop interventions aiming to enhance beliefs of physiotherapists as they can influence the prognosis of the patient [49].

The knowledge about the physiology of pain in physiotherapists was explored in this study with the NPQ. The mean score of 66% (12.6 ± 3.2) cannot be considered as good for graduated physiotherapists. Our results are higher than those observed in studies from Meeus et al. [50] and Moseley [28] with a mean of 56% (10.71 ± 3.08) and 55% (10.45 ± 3.61), respectively. Nevertheless, recent studies from Stern et al. [51] and Lane et al. [52] (using a shorter version of the NPQ (12-item) [21]) showed higher scores in physiotherapists with a mean score of 75% (9 ± 1.5) and 80% (9.6 ± 1.1), respectively. Even with these higher scores, Stern et al. concluded that physiotherapists had limitations in pain science [51]. Pain neuroscience education is an approach to reconceptualise how pain works [53]. However, this is relatively new, and one hypothesis might be that some physiotherapists in our study had not benefitted from these new insights since graduation. 

Reassurance about the pain experience is recommended in the clinical guidelines and could positively influence pain ratings, disability and limitations in movement of the patient [54]. In one study, a NPQ mean score of 90% was required for practitioners to be included and deliver pain neuroscience education [55,56]. Unfortunately, barriers to implementation in practice exist and the evolution of knowledge in pain science may not be delivered appropriately to physiotherapists and patients [57,58,59]. 

### 4.3. Clinical Vignettes

The results of the non-specific vignette developed by Rainville [23] showed that a majority of physiotherapists (63%) gave guideline-inconsistent recommendations concerning return to work and guideline-consistent recommendations when advising the patient about activities. These results are comparable to other studies which showed guideline-inconsistent recommendations concerning work in 76% [25] and 50% [47] of physiotherapists. In another study, physiotherapists gave guideline-consistent recommendations concerning work and activity (60% and 88%, respectively) [39]. In comparison, our proportion of physiotherapists giving guideline-inconsistent recommendations for work is high. These results are concerning given the fact that physiotherapists follow their patients for multiple sessions and could potentially implement unhelpful beliefs related to work, favour a worse prognosis and increase long-term disability in patients with LBP. This major difference between recommendations for activity and work could be explained for different reasons. Firstly, return to work is a topic seldom included in curricula of physiotherapists or in postgraduate training, while the opposite is true for activity recommendations. Physiotherapy curricula are mainly based on the promotion of movement and activity in patients to recover their health. Secondly, physiotherapists in Belgium and France can discuss returning to work with patients but the final decision is made by the physician. Inter-disciplinary discussions about return to work are not usually implemented in private practice. Thirdly, a clinical vignette is completely different from an interview in a clinical setting, and this could influence the given recommendations. Current clinical vignettes lack the integration of psychosocial factors. New clinical vignettes should be developed to allow a better evaluation of the situation and context by health professionals. 

Finally, half of the physiotherapists did not suspect the presence of a specific type of LBP in a clinical vignette despite clear indicators of a neurological condition that should arouse suspicion or concerns and influence clinical decision making. Only 30% of the participants detected the correct underlying specific pathology (lumbar spinal stenosis). These results are highly concerning and are similar to other studies [30,60,61] where only half of the physiotherapists recognised the specific pathology and performed clinical decision making. Even more concerningly, the cause of LBP was often wrongly attributed to the patient’s age or behavioural factors. (e.g., “*the patient don’t follow the treatment correctly*”). These results could be explained by the fact that guidelines are not consistent about which features would indicate a specific diagnosis, which may lead to confusion and inconsistency in management of patients [6]. This confusion could also have influenced physiotherapy curricula. Our results indicate that caution is needed before allowing direct access in Belgium or France. To avoid mismanagement of patients, strategies to better implement the diagnostic triage [62] and the suspicion of specific pathologies underlying musculoskeletal disorders should be developed.

### 4.4. Limitations and Strengths

This study had some limitations. The psychometric characteristics of the versions of the Back-PAQ and NPQ translated into Dutch were not studied. Moreover, the second clinical vignette (specific low back pain) was developed for the purpose of this study and was not validated. Their validity and psychometric characteristics should be analysed in future studies. To facilitate the recruitment procedure, volunteers were sought using broad advertising and accreditation points were given to physiotherapists when they finished their participation in the study. This point attribution could have biased the sample of physiotherapists recruited. Volunteer physiotherapists could be more aligned with knowledge creation and use. Nevertheless, offering “free” accreditation points may have encouraged those who generally do not follow learning opportunities. It is also important to acknowledge that this study only measures explicit attitudes and beliefs rather than implicit orientation of physiotherapists in a clinical setting. Implicit attitudes and beliefs could also greatly influence patient outcome because spontaneous and everyday clinical management is not always driven by deliberate analysis [17]. 

This study had several strengths as well. The recruitment of physiotherapists took place in two countries, and the sample of participants was large. This facilitated the gathering of up-to-date data on knowledge, attitudes and beliefs of physiotherapists in these countries. To the best of our knowledge, this is the first study to analyse the ability of physiotherapists to suspect the presence of a specific pathology causing LBP using a clinical vignette in Belgium and France. The variety of measured outcomes included in this study facilitated extensive results concerning the current knowledge, attitudes and beliefs of physiotherapists in these countries.

## 5. Conclusions

This study found that a high proportion of physiotherapists in France and Belgium were unfamiliar with guidelines related to LBP management and did not apply these in practice. This lack of knowledge concerning guidelines is reflected by beliefs that there is a strong relationship between pain and impairment, beliefs about LBP that are unhelpful and inadequate knowledge on the neurophysiology of pain. The majority of physiotherapists gave guideline-inconsistent recommendations concerning returning to work, which are known to negatively influence the prognosis of patients. Half of the physiotherapists in this study did not suspect the presence of a specific cause of LBP in a clinical vignette with features of spinal stenosis and neurological compromise. Future studies should develop and evaluate interventions aiming to better implement best practice and guideline-oriented management of LBP in physiotherapists. These future interventions should include all the aspects of clinical guidelines and the bio-psychosocial model, including important topics such as the capacity to suspect a specific cause of LBP, the evaluation of psychosocial factors and clinical tools to effectively reassure the patient about their condition.

## Figures and Tables

**Figure 1 ijerph-20-05611-f001:**
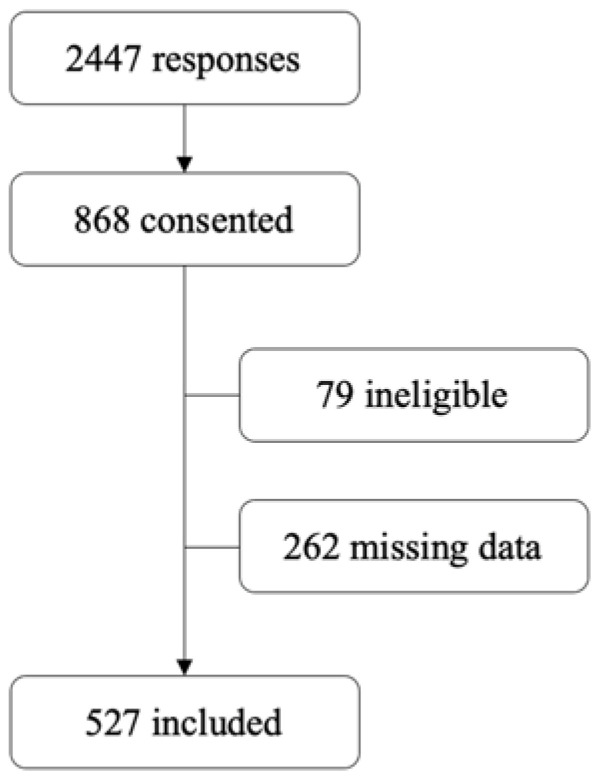
Flow diagram concerning the recruitment of physiotherapists.

**Figure 2 ijerph-20-05611-f002:**
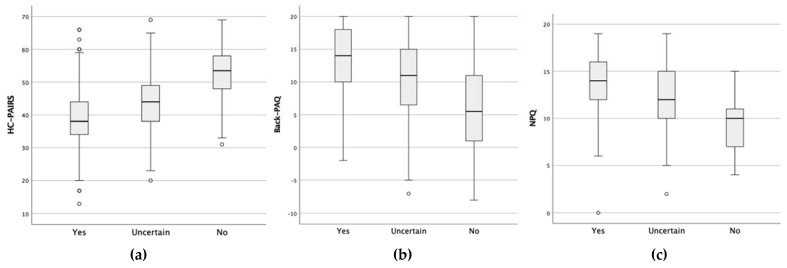
Boxplots representing the relation of the HC-PAIRS (**a**), Back-PAQ (**b**) and NPQ (**c**) according to self-reported knowledge of the guidelines.

**Table 1 ijerph-20-05611-t001:** Descriptive statistics for sociodemographic questionnaire results.

		n (%) or Mean (SD)	
**Number of participants**	Total	527	100%
	Belgium (French-speaking)	150	28%
	Belgium (Dutch-speaking)	277	53%
	France	100	19%
**Age (year)** **Age (by group)**		35	(11)
	22–32	304	58%
	33–43	105	20%
	44–54	70	13%
	55–65	42	8%
	>66	6	1%
**Gender**	Female	312	59%
	Male	215	41%
**Years of practice**		12	11.6
**Work setting (multiple answers allowed)**	Self (alone)	160	30%
	Self (in a group with same profession)	238	45%
	Multidisciplinary	103	20%
	Medical house	38	7%
**Clinical workload**	Hospital	100	19%
	Disability sector	19	4%
	50%	29	6%
	25%	12	2%
**LBP patients per month**	<5	152	29%
	5–10	42	8%
	10–15	103	19%
	15–20	193	37%
	>20	37	7%
**Self-reported knowledge of the guidelines**	Yes	197	37%
	Uncertain	312	59%
	No	18	4%
**Self-reported application of guidelines in practice**	Yes	163	31%
	Sometimes	325	62%
	No	39	7%

**Table 2 ijerph-20-05611-t002:** Descriptive statistics for the HC-PAIRS, Back-PAQ and NPQ.

	n	Median [Q1, Q3]	Minimum	Maximum
**HC-PAIRS (13–91)**	527	42 [36, 48]	13	69
**Back-PAQ (−20–20)**	527	12 [7, 16]	−8	20
**NPQ (0–19)**	527	13 [11, 15]	0	19

**Table 3 ijerph-20-05611-t003:** Scores of the Back-PAQ sub-grouped by theme.

Back-PAQ Themes	Score	Vulnerability of the Back	Relationship between Pain and Injury	Activity Participation While Experiencing Back Pain	Psychological Influences on Recovery	Prognosis of Back Pain
**Score (%)**	−2	15.6	1.4	0.6	2.9	2.9
	−1	19.2	13.1	0.7	4.3	4.3
	0	7.8	6.4	0.8	8.3	8.3
	1	16.7	20.1	10.1	44.3	44.3
	2	40.8	59	88	40.1	40.1
**Median [Q1, Q3]**		1 [−1, 2]	2 [1, 2]	2 [2, 2]	1 [1, 2]	1 [0, 2]
**Mean (SD)**		0.5 (1.5)	1.2 (1.1)	1.8 (0.5)	1.1 (0.9)	1 (1.1)

**Table 4 ijerph-20-05611-t004:** Descriptive statistics for the clinical vignette (non-specific LBP) developed by Rainville [23].

Activity	I Would Recommend This Patient to:	Score	n (%)	Work	I Would Recommend This Patient to:	Score	n (%)
**Guideline-consistent**	Not limit any activities	1	79 (15)	**Guideline-consistent**	Work full-time at full capacity	1	17 (3)
	Avoid only painful activities	2	324 (61)		Work full-time at moderate capacity	2	179 (34)
**Guideline-inconsistent**	Limit activities to moderate exertion	3	87 (17)	**Guideline-inconsistent**	Work full-time at ligth capacity	3	157 (30)
	Limit activities to light exertion	4	37 (7)		Work part-time at light capacity	4	164 (31)
	Limit all physical activities	5	0		Remain out of work	5	10 (2)
		**Total**	**527 (100)**			**Total**	**527 (100)**

**Table 5 ijerph-20-05611-t005:** Scores of the HC-PAIRS, Back-PAQ and NPQ sub-grouped by the suspicion and detection of a specific cause of LBP.

Suspicion of a Specific Cause of LBP	n (%)	HC-PAIRS Median [Q1, Q3]	Back-PAQ Median [Q1, Q3]	NPQ Median [Q1, Q3]
**No**	243 (46)	43 [36, 48]	12 [7, 15]	13 [10, 15]
**Yes**	284 (54)	41 [36, 48]	13 [7, 16]	13 [11, 15]
		***p* = 0.172**	***p* = 0.058**	***p* = 0.037**
**Detection of the Specific Cause of LBP**	n (%)	**HC-PAIRS** **Median [Q1, Q3]**	**Back-PAQ** **Median [Q1, Q3]**	**NPQ** **Median [Q1, Q3]**
**No**	369 (70)	42 [36, 49]	12 [7, 15]	12 [10, 15]
**Yes**	158 (30)	41 [35, 47]	13 [9, 17]	14 [11, 15]
		***p* = 0.081**	***p* = 0.004**	***p* = 0.004**

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
