# Peer review of "Management of Low Back Pain: Do Physiotherapists Know the Evidence-Based Guidelines?"

_ijerph, 2023, doi:10.3390/ijerph20095611_

Round 1

Reviewer 1 Report

The authors have performed a very interesting study.  Further analysis examining LBP patients per months and the three questionnaires possibly a Chi-square analysis would be very interesting.  If the physiotherapist evaluates less than 5 patients per month and is unfamiliar with the questionnaire that is much less concerning than a physiotherapist that sees 15-20 per month and is unfamiliar.  This analysis may provide some much more interesting results and further insights.

Reviewer 2 Report

Dear authors,

This is a very important area to research. I consider the ms well written and sound and interesting.  For the understanding of readers from other countries than France/Belgium I think it would be interesting to understand the context of how physiotherapists work in these countries. Are patients referred by doctor? Is it direct access to physiotherapy? What about education of physiotherapists? Post graduate courses?  Please add some short info on this to better understand the context. 

Also interesting is that 30% meet <5 patients with LBP per months which can affect the result. Also 29% work alone which also can impact on the results. Also miss out if there seem to be a difference between France and Belgium. But if you write something about the context I will better understand if the countries are alike regarding education of physiotherapists and how the health care system function regarding physiotherapy. 

Reviewer 3 Report

The manuscript appears interesting and suitable for the journal. The methodology is adequate. Publication is recommended with slight revisions.

MSD affect a high number of healthcare practitioners. Further implementation in the assessment of ergonomic knowledge is useful to prevent MSD occurrence.

I suggest only minor modifications.

In the introduction, the development ad etiopatogenesis of low back pain should be more deeply described: is it due to unbalanced posture? Which muscles are submitted to excessive strain. Report data from recent literature

Please report and discuss MSD occurrence also in other medical fields. Dentists and dental hygientists also experience a high percentage of WMSD in the low back due to forceful repetitions and akward posture. The following paper should be discussed and cited.

Ohlendorf 2020 IJERPH Prevalence of Musculoskeletal Disorders among Dentists and Dental Students in Germany

Gandolfi 2021 IJERPH Musculoskeletal disorders among italian dentists and dental hygienists

Round 2

Reviewer 2 Report

Thanks for revising according to my comments. The section that describe the context of physiotherapy in France and Belgium is better suited in beginning of Methods section. But I leave to editor to decide on that.